# Accuracy of Self-Reported Items for the Screening of Depression in the General Population

**DOI:** 10.3390/ijerph17217955

**Published:** 2020-10-29

**Authors:** Jorge Arias-de la Torre, Gemma Vilagut, Antoni Serrano-Blanco, Vicente Martín, Antonio José Molina, Jose M Valderas, Jordi Alonso

**Affiliations:** 1Department of Psychological Medicine, Division of Academic Psychiatry, Institute of Psychiatry, Psychology and Neuroscience (IoPPN), King’s College London, London SE5 8AF, UK; 2CIBER Epidemiología y Salud Pública (CIBERESP), 28029 Madrid, Spain; gvilagut@imim.es (G.V.); aserrano@pssjd.org (A.S.-B.); vicente.martin@unileon.es (V.M.); jalonso@imim.es (J.A.); 3Institute of Biomedicine (IBIOMED), University of León, 24071 León, Spain; ajmolt@unileon.es; 4Health Services Research Group, IMIM (Hospital del Mar Medical Research Institute), 08003 Barcelona, Spain; 5Institut de Recerca Sant Joan de Déu, Parc Sanitari Sant Joan de Déu, 08950 Barcelona, Spain; 6Health Services and Policy Research Group, University of Exeter Medical School, Exeter EX4 2LU, UK; j.m.valderas@exeter.ac.uk; 7Department of Experimental and Health Sciences, Pompeu Fabra University (UPF), 08002 Barcelona, Spain

**Keywords:** depression, metric properties, sensitivity and specificity, health surveys

## Abstract

Introduction: Though self-reported items (SRD, self-reported depression) are commonly used in health surveys and cohort studies, their metric properties as a depression indicator remain unclear. The aims were to evaluate the measurement properties of SRD using the Patient Health Questionnaire-8 (PHQ-8) as reference and to identify factors related to the agreement between both indicators. Methods: Data from the European Health Interview Survey in Spain in 2014/2015 (n = 22,065) were analyzed. Two indicators of depression were considered: SRD based on two items yes/no (positive: both yes), and the PHQ-8 (positive ≥ 10). Socioeconomic factors and use of health services were considered as independent variables. The prevalence of depression, sensitivity, specificity, global agreement, and positive and negative predictive values (PPV and NPV) of SRDs were evaluated using the PHQ-8 as a reference. Logistic regression models were fitted to determine factors associated with the agreement between indicators. Results: The prevalence of depression was lower when assessed with PHQ-8 (5.9%) than with SRD (7.7%). SRD sensitivity and PPV were moderate–low (52.9% and 40.4%, respectively) whereas global agreement, specificity, and NPV were high (92.7%, 95.1%, and 97.0%, respectively). Positive agreement was associated with marital status, country of birth, employment status, and social class. Negative agreement was related to all independent variables except country of birth. Conclusions: SRD items tend to overestimate the current prevalence of depression. While its use in health surveys and cohorts may be appropriate as a quick assessment of possible depression, due to their low sensitivity, its use in clinical contexts is questionable.

## 1. Introduction

Depressive disorders are amongst the most common and burdensome mental health problems and are one of the main causes of disability, dependence, and health expenditure worldwide [1]. Previous research shows differences in the prevalence of depressive disorders, even within the same populations and timeframes. It has been suggested to a large extent that these differences might be explained by differences in the indicators used to estimate prevalence [2,3].

Two of the most valid and reliable indicators used to assess depressive disorders in health surveys are the Composite International Diagnostic Interview (CIDI) and the Structured Clinical Interview for DSM-5 (SCID-5) [4]. These types of indicators could be considered as a reference for determining the prevalence of depressive disorders in health interview surveys. Nevertheless, in most cases, their use might not be feasible due to the resources required in applying them. Instead, many health surveys and cohorts rely on screening tools or patients self-reporting their depression diagnosis (SRD, self-reported depression).

Screening instruments for depression are easy to administer and interpret, generally with acceptable validity and reliability, and they cost less than structured and semi-structured interviews [5,6]. These tools determine the presence or absence of depressive symptomatology, some of which show suitable psychometric properties when compared to the clinical diagnosis or clinical interviews such as the Beck Depression Inventory (BDI) and Beck Depression Inventory-second edition (BDI-II), the Center for Epidemiologic Studies Depression Scale (CES-D), and the Hamilton Depression Rating Scale (HADRS) [5,7,8,9,10]. One of these screening tools, the use of which has substantially increased during the last years, is the Patient Health Questionnaire (PHQ) [11]. This tool has been included within health interview surveys and cohort studies both inside and outside Europe [7,12,13]. Furthermore, unlike previous tools, it is based on the standard criteria for depression as defined by DSM-IV, thus allowing us to consider it as a suitable proxy to determine depression.

Though SRD items might be efficient and have been used in different contexts such as health surveys as in the European Health Interview Survey [14] and cohort studies such as the 1970 British Cohort Study or the 1958 National Child Development Study [15,16], their metric properties remain unclear [17,18,19,20]. These indicators could be particularly useful due to the minimal burden that their use implies, which is even less than screening tools. They can also be beneficial given the simplicity of their capture, information processing, analysis, and interpretation. Therefore, determining the diagnostic performance of SRD items and their suitability for inclusion in health interview surveys and cohort studies could be particularly valuable given the potential benefits their use could imply.

Both the prevalence of depressive disorders and their association with social factors might vary depending on the depression indicator being used [17,18,21]. However, mental disorders are systematically related to sociodemographic factors such as age, gender, social class, and employment status, among other aspects, regardless of the indicator used to assess them [18,22,23]. Therefore, to obtain a clearer and broader representation of how specific factors could influence the results obtained from different indicators, an evaluation of their diagnostic performance and the factors associated with this performance is needed.

Henceforth, the aims of this study are to (a) evaluate the diagnostic accuracy of self-reported depression using the PHQ-8 as the reference standard and (b) identify factors associated with the positive and negative agreement between SRD questions and the PHQ-8.

## 2. Materials and Methods

### 2.1. Study Population

Data from the European Health Interview Survey from 2014/2015 (EHIS-2015) in Spain were used (N = 22,842). EHIS-2015 participants were a representative sample aged 15 years old or older, selected from the non-institutionalized Spanish population using a three-stage sampling method [14]. A total sample of 22,065 individuals was considered after excluding participants for whom it was impossible to determine their PHQ-8 score (n = 170; 0.7%), SRD (n = 17; <0.1%), marital status (n = 20; <0.1%), occupational social class (n = 530; 2.4%), or use of health services (n = 47; 0.2%).

### 2.2. Indicators of Depression

Self-reported depression (SRD) was evaluated from two dichotomous items (yes/no): “Have you ever suffered from depression?” and “Have you suffered from it in the past 12 months?” SRD was considered positive when the individual responded affirmatively to both items.

The Patient Health Questionnaire-8 (PHQ-8), a valid and reliable screening tool to assess the severity of depressive symptomatology based on DSM-IV criteria, was used. Due to its known reliability and validity when compared to structured clinical interviews [11,24,25], this indicator was considered in this study as the reference point in determining the presence of depression. The questionnaire is composed of 8 Likert-type items (0–3) assessing the individual’s self-reported depressive symptoms in the previous 2 weeks. A total score between 0 and 24 was calculated by adding up the responses to each item. Following previous validation studies, we used the cutoff value of 10+ for depression (sensitivity over 75%; specificity over 85%) [7,13,24,26]. Despite a temporal framework of 2 weeks, the PHQ-8 has demonstrated suitable properties to detect different depressive disorders including dysthymia, which should have a duration longer than 2 years, and to assess the lifetime prevalence of depression.

### 2.3. Sociodemographic Factors and Use of Health Services

The following variables were also taken into account in this study: gender, age in years (categorized), country of birth, level of education, marital or cohabiting status, employment status, and occupational social class, based on the current or previous job of the main breadwinner in the interviewee’s household and classified by social class from I (most advantaged) to VI (most disadvantaged). The use of health services in the previous 4 weeks by the person interviewed was also assessed.

### 2.4. Data Analysis

A descriptive analysis of the population’s characteristics and prevalence of depression from PHQ-8 and SRD items was carried out. All proportions and their respective 95% confidence intervals (95% CI) were calculated globally, as well as for each category of all the study variables. Additionally, using the diagnosis of depression with PHQ-8 as the reference standard, the following indices of test performance with their 95% CI were calculated for SRD items (0 = negative; 1 = positive): sensitivity, specificity, area under the response operative curve (AUC), global agreement, and positive and negative predictive values (PPV and NPV). Furthermore, as supplementary analysis, the sensitivity and specificity were also calculated for the PHQ-8 using the SRD as standard.

To evaluate the relationship of the explanatory factors with the agreement between PHQ-8 and SRD items, single and multiple logistic regression models were fitted. Logistic models were used due to the binary (dichotomous) nature of the dependent variables used in this study [27]. From these models, crude odds ratios (OR) and adjusted odds ratios (aOR) were obtained. Two models with SRD as the dependent variable were fitted, one including only individuals who tested positive for depression (PHQ-8 ≥ 10) to study factors associated with sensitivity and another one including only individuals who tested negative for depression (PHQ-8 score < 10) to study factors associated with specificity. These models were fitted as an approach to specificity on the one hand, i.e., considering only participants who are negative on the PHQ-8 and using SRD as the dependent variable (negative = 1 and positive = 0), and as an approach to specificity on the other hand, i.e., considering only participants who are negative on the PHQ-8 and using SRD as the dependent variable (negative = 1 and positive = 0). All multiple models were adjusted for gender, age, country of birth, level of education, marital/cohabiting status, employment status, occupational social class, and use of health services, and the absence of interactions among them was evaluated. The statistical significance for each of the variables included was calculated using Wald tests. To ensure the representativeness of the data, all analyses were performed taking the weights derived from the complex sample design into account. All analyses were performed using the statistical software STATA v.14 [28].

## 3. Results

Table 1 shows the characteristics of the study sample and the distribution of the prevalence of depression assessed by the PHQ-8 and SRD items. Of the participants, 1461 (5.9%; 95% CI: 5.5–6.3) had a PHQ-8 score equal to or greater than 10, and 1988 (7.7%; 95% CI: 7.2–8.1) had self-reported depression (SRD). Furthermore, Table 1 shows a higher prevalence of depression, both from PHQ-8 and SRD items, among older participants; women; those belonging to lower social classes; those with primary level of education or illiterate; people widowed, separated, or divorced; people who are retired or pre-retired and homemakers; those who were born in Spain; and those who reported the use of health services in the last month.

About the metric properties of SRD items when using PHQ-8 scores as the reference for identifying depression (Table 2), they were found to have a moderate–low sensitivity (52.9%, 95% CI: 52.8–53.0), a high specificity (95.1%, 95% CI: 95.1–95.2), and an AUC of 0.74 (95% CI: 0.72–0.76) (Figure 1). In addition, it was found that the global agreement was 92.7% (95% CI: 92.7–92.7), the PPV was moderate–low (40.4%, 95% CI: 40.4–40.5), and the NPV was high (97.0%, 95% CI: 97.0–97.0). Furthermore, the supplementary analysis shows that the sensitivity and specificity of the PHQ-8 considering the SRD indicator as standard were respectively 40.4%, (95% CI: 40.4–40.5) and 97.0%, (95% CI: 97.0–97.0).

Table 3 shows the results for the two logistic models for the relationship between sociodemographic variables and SRD in individuals testing either positive (model 1) or negative for depression (model 2) based on PHQ-8 scores. SRD positive status was more frequent for those who were “separated”/“divorced” (aOR: 2.41; 95% CI: 1.28–4.52 with “single” as the reference category); those with “other” employment status, including students and individuals who were unable to work (aOR: 2.51; 95% CI: 1.41–4.47; “working” as the reference); and those in social class V (aOR: 2.15; 95% CI: 1.08–4.29; I as the reference); and less frequent for those whose country of birth was Spain (aOR: 1.89; 95% CI: 1.10–3.24; “not Spain” as the reference). Regarding participants identified as not having depression by PHQ-8 (model 2), more women had a negative SRD status (aOR: 2.08; 95% CI: 1.76–2.47), those with primary education or illiterate (aOR: 1.53; 95% CI: 1.14–2.06; university studies as the reference), and with employment status, social class, respondent’s age, and use of health services in the previous month.

## 4. Discussion

The results of this study, based on a large sample representative of the Spanish general population, show that SRD assessed by two questions usually included in population health surveys and cohort studies, might overestimate the prevalence of depression when compared with estimates from the PHQ-8, a valid and reliable tool for the assessment of depression [13,25,29,30]. Furthermore, although its sensitivity and positive predictive value are moderate, its specificity and negative predictive value are high, making it a valuable tool in the context of health surveys and cohort studies.

The higher prevalence of depression found using SRD items could be caused by a possible higher likelihood for it to detect milder cases, more so than for differences in temporal frameworks. Furthermore, the moderate sensitivity and PPV of SRD items might indicate that they could be detecting a possible subclinical depression or indicate subthreshold mental health problems associated with depression. In this sense and as previous research has suggested [17,18,20], the use of SRD items might be a short, easy to use and, consequently, highly efficient screening tool to detect individuals with incipient depression or a depression-related mental morbidity. This could be highly useful in helping early detection of possible incipient cases, at the individual level and at the population level, to anticipate possible increases or decreases in depression rates, improving the quality of heath surveys as a tool to plan healthcare resources, and of cohort studies to detect depression trajectories from milder to more severe depression stages. Future studies using lower cutoff values of PHQ-8 and using SRD with other mental disorders related to depression could be valuable in determining the validity of SRD for these purposes. Additionally, as suggested by the supplementary analysis, further research using the SRD indicator as reference might be valuable to determine its relationship with the PHQ-8 as well as with other screening tools for depression and with clinical interviews (e.g., CIDI or SCID).

The high specificity and NPV of SRD items when compared with the PHQ-8 should be emphasized. In this sense, models fitted only for negatives in PHQ-8 could be an approximation for which factors could be related to sensitivity. While models for agreement for negatives show differences in all explanatory variables, barring country of birth, models for positives did not show differences by gender, level of education, age, or use of health services. As previous research has pointed out [7,9,13,31,32], depression measures such as the PHQ-8, the BDI-I and BDI-II, the CES-D, or the HDRS, might have lower diagnostic accuracy depending on the specific sociodemographic characteristics of the individuals. In the case of SRD items, the agreement for positive could be lower among those with less risk of severe depression a priori or the healthier groups, such as individuals born in Spain, those that are employed, and among individuals belonging to the higher social classes. Since estimates of sensitivity are expected to increase with increasing disease severity [33], lower agreement among individuals of these groups might be caused by the lower risk of depression in this population. Further research focused on these specific population groups, might help improve the accuracy of the SRD estimations. Furthermore, as people from the most disadvantaged extreme of the social continuum are usually not included within health interview surveys [34,35], exploring the accuracy of SRD in this segment of the population might help to better determine the accuracy of SRD to screen for depression.

Regarding the suitability of SRD for use in different contexts, the results found suggest that while this indicator might not be appropriate in clinical contexts, it could be so in health interview surveys and cohort studies [33]. In clinical contexts, tools with higher sensitivity may be fitting to correctly diagnose patients, but for health surveys or cohort studies, it might be preferable to use tools with a higher specificity [13]. In this sense, using the SRD indicator in cohort studies and particularly in health surveys could be an acceptable choice, given its short length and high specificity compared with a suitable screening tool such as PHQ-8, its low cost, ease of collection, and the possibility of using the indicator to double-check the results of valid and reliable tools like CIDI or SCID.

There are some limitations in this study that should be discussed apart from the differences in temporal frameworks between indicators addressed in their description. Though one limitation is assessing the SRD indicator using a screening tool (the PHQ-8), the cutoff value of 10+ of the PHQ-8 has shown high reliability and validity in detecting depressive disorders (including major depressive disorder and dysthymia) when compared with the CIDI, SCID, or MINI, and a suitable concordance when compared with the clinical diagnosis [7,13,24,26]. In addition, the PHQ-8 is based on the DSM-IV criteria for major depressive disorder and many researchers have shown its validity to be used in different contexts and population groups [13,25]. These characteristics makes it an indicator that could be considered as reference for screening depression at the population level. We should also highlight that SRD was assessed using 2 items. The Spanish edition of the EHIS, to be compatible with the Spanish National Health Survey, also includes an additional item about whether a physician diagnosed the disease. This item was not included in all countries involved in EHIS. Therefore, to maximize the comparability and usefulness of the results found, we decided not to consider this third item. Furthermore, this item is only applicable to those who use health services and to some extent is not self-reported, which is why we felt its exclusion from a self-reported indicator was appropriate. Finally, we should mention that the use of other statistical approaches (e.g., log-linear analyses) might be useful to better understand the relationship between the SRD and the PHQ-8 and give new information about the accuracy of the former.

## 5. Conclusions

Our study shows that the SRD tends to overestimate the current prevalence of depression and has low sensitivity. In addition to the difference in recall periods, other variables such as marital status, country of birth, employment status, and social class could explain the differences found between both measures. While the use of the SRD indicator is not adequate for assessing the prevalence of depression or for diagnostic use at the individual level in clinical contexts, its use could be valuable in health surveys and cohort studies as a quick way to screen for depression and as a measure of the need for mental health care.

## Figures and Tables

**Figure 1 ijerph-17-07955-f001:**
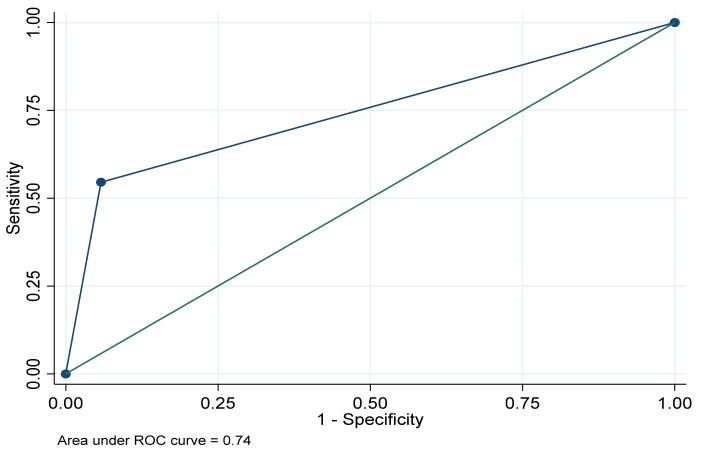
Area under the receiver operating characteristic curve (ROC) for the SRD indicator using the PHQ-8 as reference. European Health Interview Survey in Spain, 2014/2015.

**Table 1 ijerph-17-07955-t001:** Characteristics of the sample and prevalence of depression (Patient Health Questionnaire-8, PHQ-8+) and self-reported depression (SRD). European Health Interview Survey in Spain 2014/2015.

	Total Sample (n = 22,065)	PHQ-8+ (n = 1461)	SRD+ (n = 1988)
	**% (95% CI)**	**% (95% CI)**	**% (95% CI)**
**Gender**			
Men	49.2 (48.5–50.0)	3.9 (3.5–4.4)	4.7 (4.3–5.3)
Women	50.8 (50.0–51.5)	7.7 (7.1–8.4)	10.5 (9.8–11.2)
**Marital status**			
Single	26.1 (25.3–26.9)	3.6 (3.0–4.3)	4.6 (4.0–5.4)
Married or cohabiting	62.8 (62.0–63.6)	5.6 (5.1–6.1)	7.3 (6.8–7.9)
Widowed	6.9 (6.6–7.2)	15.1 (13.5–16.9)	19.2 (17.4–21.1)
Separated or divorced	4.2 (3.9–4.5)	8.5 (6.9–10.6)	12.7 (10.8–15.0)
**Level of education**			
University	18.9 (18.1–19.8)	2.4 (1.9–3.0)	3.5 (3.0–4.2)
Secondary	51.1 (50.1–52)	4.9 (4.5–5.5)	6.2 (5.6–6.8)
Primary or Illiterate	30.0 (29.0–31.0)	9.6 (8.7–10.5)	12.8 (12.0–13.8)
**Country of birth**			
Not Spain	12.9 (12.1–13.8)	4.5 (3.4–5.8)	5.5 (4.3–7.0)
Spain	87.1 (86.3–87.9)	6.1 (5.7–6.5)	8.0 (7.5–8.5)
**Working status**			
Employed	45.3 (44.4–46.3)	2.7 (2.3–3.2)	3.8 (3.3–4.3)
Unemployed	15.1 (14.4–15.8)	8.6 (7.4–9.9)	9.1 (7.9–10.4)
Retired/pre-retired	20.1 (19.4–20.7)	9.4 (8.5–10.3)	13.1 (12.1–14.2)
Homemaker	8.6 (8.1–9.0)	9.8 (8.3–11.5)	13.3 (11.6–15.2)
Other	10.9 (10.3–11.5)	5.6 (4.5–6.9)	7.7 (6.4–9.1)
**Social class**			
I	11.4 (10.7–12.1)	2.2 (1.7–3.0)	3.0 (2.4–3.8)
II	8.3 (7.8–8.8)	3.9 (3.0–5.0)	5.4 (4.4–6.7)
III	19.0 (18.4–19.8)	4.3 (3.7–51)	6.1 (5.3–7.0)
IV	14.6 (14.0–15.2)	5.2 (4.4–6.1)	8.1 (7.1–9.2)
V	32.5 (31.6–33.5)	7.2 (6.5–7.9)	9.0 (8.2–9.8)
VI	14.2 (13.4–15.0)	9.7 (8.4–11.1)	11.4 (10.1–12.9)
**Age**			
15–34 years old	26.8 (26.0–27.3)	3.0 (2.4–3.7)	2.6 (2.1–3.3)
35–49 years old	29.9 (29.5–30.7)	4.5 (3.9–5.1)	5.5 (4.9–6.3)
50–64 years old	22.7 (22.0–23.4)	7.6 (6.8–8.6)	11.1 (10.1–12.1)
≥65 years old	20.6 (20.0–21.3)	9.8 (8.9–10.8)	13.5 (12.5–14.6)
**Use of health services**			
No	65.2 (64.4–66.0)	3.3 (3.0–3.7)	
Yes	34.8 (34.0–35.7)	10.6 (9.8–11.6)	13.0 (12.1–13.9)

PHQ-8+: Patient Health Questionnaire-8 positive; SRD+: self–reported depression positive; %: percentage; 95% CI: 95% confidence interval.

**Table 2 ijerph-17-07955-t002:** Performance of self-reported depression (SRD) in identification of individuals with depression (PHQ-8 ≥ 10) in the general population. European Health Interview Survey in Spain, 2014/2015.

	Se % (95% CI)	Sp % (95% CI)	AUC (95% CI)	GA % (95% CI)	PPV % (95% CI)	NPV % (95% CI)
SRD	52.9	95.1	0.74	92.7	40.4	97.0
	(52.8–53.0)	(95.1–95.2)	(0.72–0.76)	(92.7–92.7)	(40.4–40.5)	(97.0–97.0)

95% CI: 95% confidence interval; Se: sensitivity; Sp: specificity; AUC: area under receiver operating characteristic curve (ROC); GA: global agreement; PPV: positive predictive value; NPV: negative predictive value.

**Table 3 ijerph-17-07955-t003:** Association of sample characteristics with being positive on the SRD items among PHQ-8 positives (model 1) and being negative on the SRD items among PHQ-8 negatives (model 2). European Health Interview Survey in Spain, 2014/2015.

	Model 1 (n = 1461)				Model 2 (n = 20,604)			
	**OR (95% CI)**	***p***	**aOR (95% CI)**	***p***	**OR (95% CI)**	***p* Value**	**aOR (95% CI)**	***p***
**SRD**								
**Gender**		0.057		0.265		<0.001		<0.001
Men	1.00		1.00		1.00		1.00	
Women	1.31 (0.99–1.73)		1.17 (0.89–1.55)		2.39 (2.05–2.80)		2.08 (1.76–2.47)	
**Marital status**		0.002		0.011		<0.001		0.038
Single	1.00		1.00		1.00		1.00	
Married or cohabiting	1.27 (0.86–1.87)		1.21 (0.81–1.83)		1.54 (1.24–1.91)		0.78 (0.60–1.01)	
Widowed	1.48 (0.96–2.29)		1.25 (0.75–2.08)		4.57 (3.57–5.87)		1.04 (0.76–1.40)	
Separated or divorced	2.59 (1.42–4.72)		2.41 (1.28–4.52)		2.54 (1.88–3.44)		1.07 (0.76–1.50)	
**Level of education**		0.238		0.163		<0.001		0.029
University	1.00		1.00		1.00		1.00	
Secondary	1.39 (0.84–2.31)		1.00 (0.56–1.78)		1.50 (1.18–1.90)		1.25 (0.96–1.63)	
Primary or Illiterate	1.44 (0.88–2.36)		0.77 (0.42–1.43)		3.60 (2.86–4.53)		1.53 (1.14–2.06)	
**Country of birth**		0.012		0.022		0.132		0.811
Not Spain	1.00		1.00		1.00		1.00	
Spain	1.97 (1.16–3.34)		1.89 (1.10–3.24)		1.28 (0.93–1.76)		0.96 (0.69–1.34)	
**Working status**		<0.001		0.001		<0.001		0.013
Employed	1.00		1.00		1.00		1.00	
Unemployed	1.51 (1.01–2.26)		1.43 (0.95–2.16)		1.93 (1.52–2.45)		1.73 (1.34–2.24)	
Retired/pre-retired	1.72 (1.22–2.42)		1.64 (0.94–2.88)		3.46 (2.85–4.19)		1.33 (1.00–1.79)	
Homemaker	2.03 (1.30–3.18)		1.57 (0.92–2.69)		3.27 (2.55–4.19)		1.25 (0.72–1.71)	
Other	2.90 (1.67–5.05)		2.51 (1.41–4.47)		1.53 (1.15–2.04)		2.83 (1.86–4.32)	
**Social class**		0.019		0.007		<0.001		<0.001
I	1.00		1.00		1.00		1.00	
II	1.15 (0.52–2.52)		1.01 (0.46–2.25)		1.80 (1.23–2.63)		1.65 (1.13–2.42)	
III	1.15 (0.57–2.30)		1.28 (0.61–2.67)		2.07 (1.50–2.85)		1.63 (1.16–2.29)	
IV	1.85 (0.92–3.71)		1.90 (0.90–4.02)		2.58 (1.88–3.54)		1.88 (1.33–2.65)	
V	2.01 (1.07–3.80)		2.15 (1.08–4.29)		2.42 (1.80–3.27)		1.69 (1.22–2.36)	
VI	1.53 (0.79–2.95)		1.68 (0.81–3.45)		3.45 (2.50–4.77)		2.29 (1.60–3.27)	
**Age**		0.017		0.607		<0.001		<0.001
15–34 years old	1.00		1.00		1.00		1.00	
35–49 years old	1.17 (0.69–2.00)		1.00 (0.53–1.78)		2.60 (1.83–3.71)		4.39 (2.65–7.24)	
50–64 years old	2.08 (1.25–3.48)		1.59 (0.88–2.88)		5.23 (3.72–7.35)		7.47 (4.69–11.88)	
≥65 years old	1.55 (0.93–2.48)		1.18 (0.55–2.51)		7.05 (5.04–9.86)		7.45 (4.39–12.66)	
**Use of health services**		0.080		0.307		<0.001		<0.001
No	1.00		1.00		1.00		1.00	
Yes	1.27 (0.97–1.67)		1.15 (0.88–1.52)		2.55 (2.21–2.94)		1.86 (1.60–2.17)	

OR: odds ratio; (95% CI): 95% confidence interval; aOR: odds ratio adjusted for all explanatory factors. Model 1: (approach to sensitivity) considering only participants with a positive PHQ-8 and using SRD as the dependent variable (negative = 0 and positive = 1). Model 2: (approach to specificity) considering only participants who are negative on the PHQ-8 and using SRD as the dependent variable (negative = 1 and positive = 0). All analyses were performed taking the weights derived from the complex sampling method into account.

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
