# Peer review of "Accuracy of Self-Reported Items for the Screening of Depression in the General Population"

_ijerph, 2020, doi:10.3390/ijerph17217955_

Round 1

Reviewer 1 Report

The motivation is relevant since the self-reported depression feeling matters. Also, the study uses a representative sample in Spain. The methodology is well described.

I only think that the paper can describe better in the methodology when it is  written that uses bivariate and multivariate logistic regression models. In fact it is not bi or multivariare since the response variable has dimension 1. So it uses multiple, not multivariate regression. Also, it is important to cite here which are the two and several explanatory variables. 

Another minor detail is to replace "for study factors associated with" with "to study..." in lines 127 and 128 on page 3.

Author Response

Accuracy of self-reported items to screening for depression in the general population [Manuscript ID: ijerph-933540]

Reviewer #1:

The motivation is relevant since the self-reported depression feeling matters. Also, the study uses a representative sample in Spain. The methodology is well described.

I only think that the paper can describe better in the methodology when it is written that uses bivariate and multivariate logistic regression models. In fact, it is not bi or multivariate since the response variable has dimension 1. So, it uses multiple, not multivariate regression. Also, it is important to cite here which are the two and several explanatory variables.

R: Many thanks you for the positive evaluation of our article. We have changed bivariable and multivariable models by single and multiple logistic regression models. In addition, we have included a more detailed information about how models were fitted, and the variables included within them.

Changes in text:

  • Please see lines 134 to 148 of materials and methods:

“To evaluate the relationship of the explanatory factors with the agreement between PHQ-8 and SRD items, single and multiple logistic regression models were fitted. Logistic models were used due to the binary (dichotomous) nature of the dependent variables used in this study. From these models, crude Odds Ratios (OR) and adjusted Odds Ratios (aOR) were obtained. Two models with SRD as the dependent variable were fitted, one including only individuals who tested positive for depression (PHQ-8≥10) to study factors associated with sensitivity and another one including only individuals who tested negative for depression (PHQ-8 score <10) for study factors associated with specificity. These models were fitted as an approach to specificity on the one hand, i.e. considering only participants who are negative on the PHQ-8 and using SRD as the dependent variable (negative=1 and positive=0), and as an approach to specificity on the other hand, i.e. considering only participants who are negative on the PHQ-8 and using SRD as the dependent variable (negative=1 and positive=0). All multiple models were adjusted for gender, age, country of birth, level of education, marital/cohabiting status, employment status, occupational social class, and use of health services and the absence of interactions among them was evaluated”.

Another minor detail is to replace "for study factors associated with" with "to study..." in lines 127 and 128 on page 3.

R: Thank you for this comment and apologies for the mistake. We have now corrected the phrase.

Changes in text:

  • Please see line 139 of materials and methods:

“Two models with SRD as the dependent variable were fitted, one including only individuals who tested positive for depression (PHQ-8≥10) to study factors associated with sensitivity and another one including only individuals who tested negative for depression (PHQ-8 score <10) for study factors associated with specificity”.

Reviewer 2 Report

The article is of great scientific interest, it is coherent, serious and well posed.

The citations are adequate, although it is estimated that some well-known authors such as Beck are missing, as well as the mention of instruments that have previously measured depression. In this sense, I think that the instrument could be associated with a theoretical model in an explicit way in which the differences and similarities that it has with others are raised and justified.

The method is adequate, the results are presented very well from the tables, which are explained in an organized way. The results are complete and consistent with what is stated in the text. The conclusions are measured and also raise the limitations of the study, the references are current and pertinent, although, as said, some references are missing from the discussion.

Author Response

Accuracy of self-reported items to screening for depression in the general population [Manuscript ID: ijerph-933540]

Reviewer #2:

The article is of great scientific interest, it is coherent, serious and well posed.

The citations are adequate, although it is estimated that some well-known authors such as Beck are missing, as well as the mention of instruments that have previously measured depression. In this sense, I think that the instrument could be associated with a theoretical model in an explicit way in which the differences and similarities that it has with others are raised and justified.

The method is adequate, the results are presented very well from the tables, which are explained in an organized way. The results are complete and consistent with what is stated in the text. The conclusions are measured and also raise the limitations of the study, the references are current and pertinent, although, as said, some references are missing from the discussion.

R: Thank you for the positive evaluation of our manuscript. We have now mentioned some other questionnaires used for the assessment of depression such as the Beck Depression Inventory (BDI) and Beck Depression Inventory-second edition (BDI-II), the Center for Epidemiologic Studies Depression Scale (CES-D) and the Hamilton Depression Rating Scale (HADRS). Furthermore, we have included a better justification of the theoretical model associated to the PHQ (based in the DSM-IV criteria) comparing it with those theoretical models associated to the other questionnaires. Additionally, we have included some relevant references.

Changes in text:

  • Please see lines 60 to 72 of the introduction:

“Screening instruments for depression are easy to administer and interpret, generally with acceptable validity and reliability, and they cost less than structured and semi-structured interviews (Gelaye et al., 2014; Gilbody et al., 2005). These tools determine the presence or absence of depressive symptomatology, some of which show suitable psychometric properties when compared to the clinical diagnosis or clinical interviews such as the Beck Depression Inventory (BDI) and Beck Depression Inventory-second edition (BDI-II), the Center for Epidemiologic Studies Depression Scale (CES-D) and the Hamilton Depression Rating Scale (HADRS). (Bagby et al., 2004; Beck et al., 1996; Gilbody et al., 2005; Levis et al., 2019; Vilagut et al., 2016). One of these screening tools that it use has substantially increase during the last years is the Patient Health Questionnaire (PHQ) (Kroenke et al., 2001). This tool has been included within health interview surveys and cohort studies both inside and outside Europe (Levis et al., 2019; Moriarty et al., 2015; Tomitaka et al., 2018). Furthermore, in difference with previous tools, is based on the standard criteria for depression as defined by DSM-IV, thus allowing consider it as a suitable proxy to determine it”.

  • Please see lines 214 to 217 of discussion:

“As previous research has pointed out (Ayuso-Mateos et al., 2001; Beck et al., 1988; Levis et al., 2019; Moriarty et al., 2015; Vilagut et al., 2016), depression measures such as the PHQ-8, the BDI-I and BDI-II, the CES-D or the HDRS, might have lower diagnostic accuracy depending on the specific socio-demographic characteristics of the individuals”.

  • Please see the following references:
  1. “Ayuso-Mateos, J.L., Vázques-Barquero, J.L., Dowrick, C., Lehtinen, V., Dalgard, O.S., Casey, P., Wilkinson, C., Lasa, L., Page, H., Dunn, G., Wilkinson, G., Ballesteros, J., Birkbeck, G., Børve, T., Costello, M., Cuijpers, P., Davies, I., Diez-Manrique, J.F., Fenlon, N., Finne, M., Ford, F., Gaite, L., Gomez del Barrio, A., Hayes, C., Herrán, A., Horgan, A., Koffert, T., Jones, N., Lehtilä, M., McDonough, C., Michalak, E., Murphy, C., Nevra, A., Nummelin, T., Sohlman, B., 2001. Depressive disorders in Europe: Prevalence figures from the ODIN study. Br. J. Psychiatry. https://doi.org/10.1192/bjp.179.4.308
  2. Bagby, R.M., Ryder, A.G., Schuller, D.R., Marshall, M.B., 2004. The Hamilton Depression Rating Scale: Has the gold standard become a lead weight? Am. J. Psychiatry. https://doi.org/10.1176/appi.ajp.161.12.2163
  3. Beck, A.T., Steer, R.A., Ball, R., Ranieri, W.F., 1996. Comparison of Beck depression inventories -IA and -II in psychiatric outpatients. J. Pers. Assess. https://doi.org/10.1207/s15327752jpa6703_13
  4. Beck, A.T., Steer, R.A., Carbin, M.G., 1988. Psychometric properties of the Beck Depression Inventory: Twenty-five years of evaluation. Clin. Psychol. Rev. https://doi.org/10.1016/0272-7358(88)90050-5
  5. Gilbody, S., Sheldon, T., House, A., 2008. Screening and case-finding instruments for depression: a meta-analysis. Can. Med. Assoc. J. 178, 997–1003. https://doi.org/10.1503/cmaj.070281
  6. Levis, B., Benedetti, A., Thombs, B.D., 2019. Accuracy of Patient Health Questionnaire-9 (PHQ-9) for screening to detect major depression: individual participant data meta-analysis. BMJ l1476. https://doi.org/10.1136/bmj.l1476
  7. Vilagut, G., Forero, C.G., Barbaglia, G., Alonso, J., 2016. Screening for Depression in the General Population with the Center for Epidemiologic Studies Depression (CES-D): A Systematic Review with Meta-Analysis. PLoS One 11, e0155431. https://doi.org/10.1371/journal.pone.0155431”.

Reviewer 3 Report

Frankly, I don't think this is enough for a paper. Seven persons have collaborated on a STATA analysis of data but it seems that they are not responsible for the data or any of the instruments, and they haven't even done the minimal job of equating the two tests. This reflects in the conclusion: the two tests are not equivalent, but the short one has the advantage of being short.

When we have two tests with cut points defined independently and arbitrarily, equivalence can happen only by chance. The short test is extremely granular, having only four possible scores. So one would take the unusual step of equating the established test (24 possible scores) to the new one. Take a look, for example, at this page in our blog: https://dexterities.netlify.app/2018/05/25/equating-a-pass-fail-score-with-dexter/

Of course, one would not expect the authors to read our blog or know about work that has not been published in a major journal. But the originality of our approach has to do with equating with data missing by design -- for compete data, it is not so original. If the authors had performed an equating, they could have come up with at least the modest result that having two positive responses on the SRD is roughly equivalent to that many points on the PHQ-8.

I am not very convinced by the logistic regressions either. Most of the demographic variables are practically uncorrelated or weakly or trivially correlated. Why not tackle each of them with a loglinear analysis of the three-way table including the two alternative measures and the background variable?

Author Response

Accuracy of self-reported items to screening for depression in the general population [Manuscript ID: ijerph-933540]

Reviewer #3:

Frankly, I don't think this is enough for a paper. Seven persons have collaborated on a STATA analysis of data but it seems that they are not responsible for the data or any of the instruments, and they haven't even done the minimal job of equating the two tests. This reflects in the conclusion: the two tests are not equivalent, but the short one has the advantage of being short.

R: Thank you for raising these concerns. We have now changed the manuscript to address some of them trying to be as accurate as possible and, furthermore, to clarify the rationale for performing this study.

When we have two tests with cut points defined independently and arbitrarily, equivalence can happen only by chance. The short test is extremely granular, having only four possible scores. So, one would take the unusual step of equating the established test (24 possible scores) to the new one. Take a look, for example, at this page in our blog: https://dexterities.netlify.app/2018/05/25/equating-a-pass-fail-score-with-dexter/

Of course, one would not expect the authors to read our blog or know about work that has not been published in a major journal. But the originality of our approach has to do with equating with data missing by design -- for compete data, it is not so original. If the authors had performed an equating, they could have come up with at least the modest result that having two positive responses on the SRD is roughly equivalent to that many points on the PHQ-8.

R: We thank the Reviewer for this comment. We have read the blog post with attention and made changes in the manuscript accordingly. It should be noted that, as all participants have answered both tests, our data could be considered complete data (using the blog’s terminology). Therefore, equating with data missing by design is not possible.

About the equating proposed, in fact, we had performed it but not for the PHQ-8 but for the SRD items. The decision of taking the PHQ-8 as reference was based on its widely proved validity and reliability. The PHQ-8 questionnaire and the selected cut-off score (10+) have shown a very good balance between sensitivity and specificity maximizing the area under ROC in different contexts and with different populations as shown in the meta-analyses published about it (Diez-Quevedo et al., 2001; Levis et al., 2019; Manea et al., 2015; Moriarty et al., 2015). As the accuracy of SRD questions was not proved, we deem that their use as a reference might be less suitable than the use of a valid and reliable tool, as is the PHQ-8.

Despite this, considering the comments done by the Reviewer and the equating approach proposed on his blog post, we have calculated the sensitivity and specificity for the PHQ-8 using the SRD as standard as supplementary analysis. Furthermore, we have now included a figure (Figure 1) showing the ROC curve for the SRD indicator to add more information about the equating performed. Finally, we have included new information supporting the validity of the use as reference the PHQ-8 in the methods section, and some discussion about the questions raised related to the approach adopted and its limitations.

Changes in text:

  • Please see lines 67 to 72 of introduction:

“One of these screening tools that its use has substantially increase during the last years is the Patient Health Questionnaire (PHQ) (Kroenke et al., 2001). This tool has been included within health interview surveys and cohort studies both inside and outside Europe (Levis et al., 2019; Moriarty et al., 2015; Tomitaka et al., 2018). Furthermore, in difference with previous tools, is based on the standard criteria for depression as defined by DSM-IV, thus allowing consider it as a suitable proxy to determine it”.

  • Please see lines 107 to 109 of materials and methods:

“Due to their known reliability and validity when compared to structured clinical interviews (Diez-Quevedo et al., 2001; Kroenke et al., 2001; Wu et al., 2019)”.

  • Please see lines 132 and 133 of materials and methods:

“Furthermore, as supplementary analysis, the sensitivity and specificity were also calculated for the PHQ-8 using the SRD as standard”.

  • Please see lines 166 and 168 of results:

“Furthermore, the supplementary analysis shows that the sensitivity and specificity of the PHQ-8 considering the SRD indicator as standard were respectively 40.4%, (95% CI: 40.4-40.5) and 97.0%, (95% CI: 97.0-97.0)”.

  • Please see Figure 1.
  • Please see lines 186 to 190 of discussion:

“The results of this study, based on a large sample representative of the Spanish general population, show that SRD assessed by two questions usually included in population health surveys and cohort studies, might overestimate the prevalence of depression when compared with estimates from the PHQ-8, a valid and reliable tool for the assessment of depression (Gilbody et al., 2008; Manea et al., 2012; Moriarty et al., 2015; Wu et al., 2019)”.

  • Please see lines 206 to 208 of discussion:

“Additionally, as suggested by the supplementary analysis, further research using the SRD indicator as reference might be valuable to determine its relationship with the PHQ-8 as well as with other screening tools for depression and with clinical interviews (e.g. CIDI or SCID).”.

  • Please see lines 236 to 245 of discussion:

“There are some limitations in this study that should be discussed apart from the differences in temporal frameworks between indicators addressed in their description. Though one limitation is assessing the SRD indicator using a screening tool (the PHQ-8), the cut-off value of 10+ of the PHQ-8 has shown high reliability and validity in detecting depressive disorders (including Major Depressive Disorder and dysthymia) when compared with the CIDI, SCID or MINI, and a suitable concordance when compared with the clinical diagnosis. (Diez-Quevedo et al., 2001; Levis et al., 2019; Manea et al., 2015; Moriarty et al., 2015). In addition, the PHQ-8 is based on the DSM-IV criteria for Major Depressive Disorder and have shown its validity to be used in different contexts and population groups (Moriarty et al., 2015; Wu et al., 2019). These characteristics makes it an indicator that could be considered as reference for screening depression at population level.

  • Please see lines 259 to 261 of conclusions:

“While the use the SRD indicator is not adequate for assessing the prevalence of depression nor for diagnostic use at the individual level in clinical contexts, its use could be valuable in health surveys and cohort studies as a quick way to screen for depression and as a measure the need for mental health care”. 

  • Please see the following references:
  1. “Wu, Y., Levis, B., Riehm, K.E., Saadat, N., Levis, A.W., Azar, M., Rice, D.B., Boruff, J., Cuijpers, P., Gilbody, S., Ioannidis, J.P.A., Kloda, L.A., McMillan, D., Patten, S.B., Shrier, I., Ziegelstein, R.C., Akena, D.H., Arroll, B., Ayalon, L., Baradaran, H.R., Baron, M., Bombardier, C.H., Butterworth, P., Carter, G., Chagas, M.H., Chan, J.C.N., Cholera, R., Conwell, Y., de Man-van Ginkel, J.M., Fann, J.R., Fischer, F.H., Fung, D., Gelaye, B., Goodyear-Smith, F., Greeno, C.G., Hall, B.J., Harrison, P.A., Härter, M., Hegerl, U., Hides, L., Hobfoll, S.E., Hudson, M., Hyphantis, T., Inagaki, M., Jetté, N., Khamseh, M.E., Kiely, K.M., Kwan, Y., Lamers, F., Liu, S.-I., Lotrakul, M., Loureiro, S.R., Löwe, B., McGuire, A., Mohd-Sidik, S., Munhoz, T.N., Muramatsu, K., Osório, F.L., Patel, V., Pence, B.W., Persoons, P., Picardi, A., Reuter, K., Rooney, A.G., Santos, I.S., Shaaban, J., Sidebottom, A., Simning, A., Stafford, L., Sung, S., Tan, P.L.L., Turner, A., van Weert, H.C., White, J., Whooley, M.A., Winkley, K., Yamada, M., Benedetti, A., Thombs, B.D., 2019. Equivalency of the diagnostic accuracy of the PHQ-8 and PHQ-9: a systematic review and individual participant data meta-analysis. Psychol. Med. 1–13. https://doi.org/10.1017/S0033291719001314
  2. Diez-Quevedo, C., Rangil, T., Sanchez-Planell, L., Kroenke, K., Spitzer, R.L., 2001. Validation and utility of the patient health questionnaire in diagnosing mental disorders in 1003 general hospital Spanish inpatients. Psychosom. Med. 63, 679–86
  3. Gilbody, S., Sheldon, T., House, A., 2008. Screening and case-finding instruments for depression: a meta-analysis. Can. Med. Assoc. J. 178, 997–1003. https://doi.org/10.1503/cmaj.070281
  4. Kroenke, K., Spitzer, R.L., Williams, J.B., 2001. The PHQ-9: validity of a brief depression severity measure. J. Gen. Intern. Med. 16, 606–13.
  5. Levis, B., Benedetti, A., Thombs, B.D., 2019. Accuracy of Patient Health Questionnaire-9 (PHQ-9) for screening to detect major depression: individual participant data meta-analysis. BMJ l1476. https://doi.org/10.1136/bmj.l1476
  6. Manea, L., Gilbody, S., McMillan, D., 2015. A diagnostic meta-analysis of the Patient Health Questionnaire-9 (PHQ-9) algorithm scoring method as a screen for depression. Gen. Hosp. Psychiatry 37, 67–75. https://doi.org/10.1016/j.genhosppsych.2014.09.009
  7. Manea, L., Gilbody, S., McMillan, D., 2012. Optimal cut-off score for diagnosing depression with the Patient Health Questionnaire (PHQ-9): a meta-analysis. Can. Med. Assoc. J. 184, E191–E196. https://doi.org/10.1503/cmaj.110829
  8. Moriarty, A.S., Gilbody, S., McMillan, D., Manea, L., 2015. Screening and case finding for major depressive disorder using the Patient Health Questionnaire (PHQ-9): a meta-analysis. Gen. Hosp. Psychiatry 37, 567–576. https://doi.org/10.1016/j.genhosppsych.2015.06.012”.

I am not very convinced by the logistic regressions either. Most of the demographic variables are practically uncorrelated or weakly or trivially correlated. Why not tackle each of them with a loglinear analysis of the three-way table including the two alternative measures and the background variable?

R: Thank you for raising this concern. We want to highlight that one of the main assumptions of the logistic model is the absence of multicollinearity. This situation usually appears when the variables included as predictors within the model are correlated. Therefore, we consider that (among other reasons e.g. the dichotomous nature of the dependent variable) the use of a logistic model is justified in absence of correlation between the demographic variables (please see the reference: Szklo and Nieto, 2014). Despite this, we have now included more information to justify its use and a suggestion for further research to explore other analytical approaches such as a log-linear analysis.

Changes in text:

  • Please see lines 136 and 137 of materials and methods:

“Logistic models were used due to the binary (dichotomous) nature of the dependent variables used in this study (Szklo and Nieto, 2014)”.

  • Please see lines 252 to 254 of discussion:

“Finally, we should mention that the use of other statistical approaches (e.g. log-linear analyses) might be useful to better understand the relationship between the SRD and the PHQ-8 and give new information about the accuracy of the former”.

  • Please see the following references:
  1. “Szklo, M., Nieto, J., 2014. Epidemiology, Beyond the Basics (3rd Edition), Sudbury, MA: Jones & Bartlett Publishers.”.
